# Near-Infrared Stimulation in Psychiatry Disorders: A Systematic Review of Efficacy and Biological Mechanisms

**DOI:** 10.3390/neurosci6010026

**Published:** 2025-03-17

**Authors:** Joanna Woźniak, Michał Pazdrak, Ada Domanasiewicz, Jakub Kaźmierski

**Affiliations:** Department of Old Age Psychiatry and Psychotic Disorders, Medical University of Lodz, 92-216 Lodz, Poland; michal.pazdrak@stud.umed.lodz.pl (M.P.); ada.domanasiewicz@stud.umed.lodz.pl (A.D.)

**Keywords:** photobiomodulation, dementia, Parkinson’s disease, depression, anxiety, therapy

## Abstract

Background: Photobiomodulation (PBM), also referred to as low-level light therapy (LLLT), is an emerging non-pharmacological approach. This treatment is considered low-risk, cost-effective, and non-invasive, utilizing near-infrared light (NIR). The purpose of this paper is to explore the underlying mechanism of action and conduct a systematic review of pre-clinical and clinical research on the use of PBM for psychiatric disorders. Methods: A search on the PubMed, Cochrane Library, and EMBASE databases was performed on 18 and 26 January 2024. Publications focused on PBM treatment in psychiatric disorders such as major depressive disorder, general anxiety disorder, dementia, Parkinson’s disease, traumatic brain injury, schizophrenia, and sexual disfunctions were included (n = 23). Results: Near-infrared stimulation is presented as an effective method, comparable to psychopharmacological treatment. The primary suggested mechanism for PBM is the stimulation of mitochondrial metabolism following the absorption of NIR energy by cytochrome C oxidase. Because of the method of implementation, which omits the liver metabolism of cytochrome P450, PMB is recognized as safe as it does not interact with other drugs. Limitations: Clinical studies vary in terms of population and treatment parameters, and most do not include a suitable control group. Conclusions: Preliminary results support the potential of NIR stimulation as a novel and innovative treatment for psychiatry. Further studies are needed to estimate the proper protocols of parameters singly for any disease.

## 1. Introduction

According to data presented by the Institute of Health Metrics and Evaluation, in 2019, one in every eight people was suffering from a mental disease worldwide [1,2]. The prevalence differs depending on the kind of mental problem. Depressive and anxiety disorders were the most common ones. Regarding the 2014 report of the Adult Psychiatric Morbidity Survey (APMS) in England, around one adult in six (15.7%) was identified with symptoms of common mental disorders, including major depressive disorder and anxiety disorders [3,4,5,6].

Dementia is another important mental health issue, in most cases due to Alzheimer’s disease (AD). AD affects 30 million elderly people worldwide. The WHO predicts that the total number of people suffering from dementia (including AD) in 2030 will be 82 million, rising to 152 million by 2050 [3].

Despite the availability of advanced pharmacological treatment in psychiatry and the discovery of novel molecules, the efficacy of drug-related treatment is still limited. Many patients experience low tolerance to medications, side effects, only partial recovery, or even drug resistance. For this reason, researchers are continuously looking for new, non-pharmacological methods.

One of the promising non-pharmacological treatment methods is photobiomodulation (PBM). PBM, also called low-level laser therapy, is based on non-invasive brain stimulation with near-infrared (NIR) light of 780 nm to 1mm wavelengths, generated by an NIR diode with a frequency of 40 Hz and a duty cycle of 50%. NIR has well-evidenced penetration to the targeted tissues, approximately 40 mm of depth. The PBM technique can be used for both transcranial and intranasal stimulation of the central nervous system. In the case of intranasal near-infrared stimulation (iNIRS), the target area is the DMN (default mode network), the resting state network responsible for maintaining baseline activity of cognitive functions during rest periods as well as metabolic homeostasis of the central nervous system. Transcranial near-infrared stimulation targets various regions of the brain depending on the specific therapy and goals. Commonly, it is used to stimulate cortical regions. Research has identified cytochrome c oxidase, also known as transport complex IV, as the key photoceptor for infrared photonic stimulation. This light absorption triggers photoexcitation within the mitochondria, leading to a cascade of cellular events that stimulates the production of ATP, regulates levels of signaling molecules such as Ca^2+^ and reactive oxygen species (ROS), and releases nitric oxide from cytochrome c oxidase [7].

The main risks of near-infrared stimulation are possible side effects. Some individuals have reported minor side effects such as headaches, fatigue, eye strain, or the feeling of warmth in the place of stimulation.

Due to its action, PBM could be used as a potential therapeutic strategy for neurodegenerative and various psychiatric diseases and is already exploited in medicine for treatment. The aim of this paper is to perform a systematic review of pre-clinical and clinical studies on PBM in the most common psychiatric conditions, such as depression, anxiety, cognitive function decline in dementia due to Alzheimer’s disease/Parkinson’s disease/traumatic brain injury, and psychotic disorders. The purpose is to show the positive effect of PBM on the diseases mentioned above.

## 2. Methods

To identify pre-clinical and clinical studies on PBM for psychiatric conditions, we performed a systematic online review in line with Preferred Reporting Items for Systematic Reviews and Meta-Analyses statement (PRISMA) guidelines [8]. We searched PubMed, Cochrane Library, and EMBASE on the 18 and 26 January 2024 and on the 2 and 6 September 2024, using the following keywords: (“Photobiomodulation” OR “LLLT” OR “Low-level laser therapy”) AND (“depression” OR “anxiety” OR “Dementia” OR “Cognition” OR “Brain injury” OR “Parkinson’s Disease” OR “Sexual Disfunctions” OR “Obsessive-compulsive disorder” OR “General anxiety disorder” OR “Traumatic Brain Injury” OR “Schizophrenia”).

Publications were then selected if they had been published in a peer-reviewed journal, described subjects with a diagnosis of the following mental conditions (depression, anxiety, dementia, cognitive impairment, Parkinson’s disease, and sexual dysfunctions), and the treatment of photobiomodulation was used (criteria one). Of the studies meeting these criteria, a further selection was made of those reporting original research—randomized or non-randomized controlled trials and open studies excluding single case reports (criteria two).

There were no search restrictions in terms of language or date of publication. Abstracts presented at conferences were excluded. Studies based on animal models were also excluded. Studies using infrared light other than the NIR spectrum were also excluded, since higher wavelengths are mechanistically different as they act mostly by inducing a thermal effect instead of modulating cytochrome C oxidase (CCO) activity. 

There were no publications related to Obsessive Compulsive Disorder (OCD).

## 3. Results

After duplicates were removed, there were 299 records identified with the search terms, of which 92 met criteria one. After criteria two was applied, 32 articles were finally included (Figure 1). The following sections discuss the studies that presented original data on the use of PBM for dementia and cognitive impairment (11), depression (4), anxiety (2), brain injury (4), Parkinson’s disease (2), sexual disfunction (1) traumatic brain injury (6), and psychotic disorders (1).

### 3.1. PBM in Dementia

The vast majority of conducted research studies related to usage of PBM referred to cognitive decline and dementia, especially dementia due to Alzheimer’s disease.

D.W. Barret and F. Gonzalez-Lima [9] conducted the first controlled study demonstrating the beneficial effect of photobiomodulation on cognitive and affective issues in people. They collected data from individuals (n = 40, 20 in treated group and 20 in control group) before and after the procedure of transcranial PBM, 1064 nm wavelength, targeting the frontal cortex. The results indicated statistically significant improvement in the Positive and Negative Affect Schedule (PANAS) (*p* = 0.043) in overall affect scores. The treated group showed significant improvement in reaction time in a sustained vigilance test (*p* = 0.047) and in the number of correct trials in the within-subject change (*p* = 0.014) tested by the Psychomotor Vigilance Task (PVT). There were also significant positive effects in the pre–post data in the treated group on memory retrieval latency (*p* = 0.021), response times (*p* < 0.001), and number of correct responses (*p* = 0.012) in the Delayed Match-to-Sample task (DMS).

According to other studies [10,11], patients treated with PBM show significant improvement in Mini Mental State Examination (MMSE) scores. In both studies, statistical significance achieved *p* < 0.001 in the treated group. PBM active groups also revealed a remarkable increase in reaction time, the percentage of correct responses, average story recall scores in the Logical Memory Test, Boston Naming Test scores, and delayed recall subtest scores in the Auditory Verbal Learning Test (*p* < 0.001; *p* < 0.015; *p* < 0.03; *p* = 0.02; *p* = 0.015, respectively).

Berman et al. [12] conducted a pilot, double-blind, randomized, placebo-controlled trial on a group of 11 participants diagnosed with MCI or AD using an NIR 1072 nm light helmet. Preliminary results indicated improvement in executive functions and retrieval memory, *p* = 0.03 and *p* = 0.035, respectively, but ultimately there was no statistical significance due to the insufficient number of subjects.

Saltmarche et al., 2017 [13] reported a five-case-series study of dementia patients treated with transcranial combined with intranasal PBM. Despite the small number of subjects and absence of control group, statistical importance was revealed for the MMSE *p* < 0.003 and ADAS-Cog *p* < 0.023.

Apart from the transcranial method of stimulation, there are some data indicating that indirect PBS stimulation through targeting other parts of the body, e.g., the abdomen, can also induce the same positive, neuroprotective effect on dementia patients, but the mechanism is unknown. Blivet et al. [14] performed a double-blind, randomized, sham-controlled trial on mild to moderate AD patients (n = 53, 27 in treated group, 26 in control group) using an innovative brain–gut PBM therapy. After the procedure, the results showed a decrease in the ADAS-Cog comprehension sub-score (*p* = 0.029) and execution times in the Trail Making Test B (*p* = 0.012). The change in the forward verbal span score also demonstrated statistically significant improvement (*p* = 0.033).

Alzheimer’s disease as a predominant neurodegenerative disorder occurs mostly among elderly people because of the progressive course of cognitive function decline. The single cause of Alzheimer’s disease is still unclear, but coincidence with some other conditions is observed. One of the conditions that correlates with AD is anemia, defined as a lowered hemoglobin concentration (<12 g/dL and <13 g/dL in women and men, respectively). For this reason, by increasing cerebral metabolic activity and blood flow via antioxidant pathways, Nagy et al. [15] used photobiomoduliation therapy combined with aerobic exercises to conduct a randomized controlled trial on a group of anemic elderly Alzheimer’s patients (n = 60). There were statistically significant changes (*p* < 0.001) in all examined assessments and parameters—the Montreal Cognitive Assessment scale, Quality of Life for AD, Berg Balance Assessment scale, hemoglobin level, Body Mass Index, and waist-to-hip ratio.

Despite the photochemical mechanism of PBM, another considered positive action is the modulation of cerebral electrical activity, especially at high-frequency oscillations. The origin of abnormalities in neural oscillations in AD patients was proved for both slow and fast ones due to β-42 amyloid formation. Gamma frequency activity is believed to arise from the interplay between cortical inhibitory interneurons and excitatory principal neurons through high rates of interneuron firing. Data collected by Spera et al. [16] confirm that continuous transcranial PBM treatment significantly influenced open-eye EEG rhythms on account of the increasing power spectral density (PSD) for gamma and beta frequency bands (*p* < 0.02 and *p* < 0.03, respectively). There was also significance for gamma frequencies for resting-state recordings with closed eyes (*p* < 0.015). However, there were no significant differences in pre–post results for cerebral blood flow or the cognitive 2-back task.

### 3.2. PBM in Depression and Anxiety

The comorbidity of depression and anxiety among psychiatric disorders occurs more often than expected. According to The World Health Organization World Mental Health Survey published in 2015, 45.7% of responders with lifetime major depressive disorder experienced one or more lifetime anxiety disorders [1].

Kerppers et al. [17] conducted a study in a group of university students with depression and anxiety (n = 22). Participants were evaluated with the Hospital Anxiety and Depression Scale (HADS), visual memory tests (faces and drawings), and the grip strength test. Researchers divided subjects into eight groups non-randomly: placebo or photobiomodulation/anxiety or depression/before or after procedure. Individuals were treated with NIR 945 nm wavelengths for 1 min 25 s daily for 30 consecutive days. Results clearly showed significant improvement for the treated anxiety and depression groups simultaneously. The scores in the HADS anxiety subscale decreased from 10.75 ± 2.49 to 6.66 ± 2.50 (*p* = 0.015) points before and after PBM, respectively. The same differences in scores were observed in the HADS depression subscale, with statistical significance (*p* = 0.0125). Moreover, there were also significant improvements in results among the depression group after the active PBM course versus the depression group after placebo. Other findings for both memory tests indicated positive significant changes for PBM treatment for depression subjects.

Another relevant study investigated the antidepressant effect of transcranial photobiomodulation in subjects diagnosed with major depressive disorder (MDD). Cassano et al. [18] conducted double-blind, sham-controlled pilot trial, ELATED-2, using an NIR PBM device of 823nm wavelength twice a week for 8 weeks. Twenty-one subjects met the inclusion criteria initially, but only thirteen followed the entire treatment period and follow-up. Individuals were assessed with the Hamilton Depression Rating Scale, where the mean change in scores indicated substantial improvement among treatment completers vs. sham (−15.7 ± 4.41; −6.1 ± 7.86; *p* = 0.031, respectively). The antidepressant effect of PBM was also confirmed via the self-reported Quick Inventory of Depressive Symptomatology.

Based on previous findings on MDD, Maiello et al. [19] investigated the anxiolytic effect of photobiomodulation in a group of subjects (n = 15) with primary generalized anxiety disorder (GAD). Each patient received 8 weeks of transcranial PBM treatment, once daily for 15 min in the first week and 20 min thereafter. The achieved results showed the significant difference between before–after scores in the Clinical Global Impressions—Severity scale (4.5 ± 0.52 to 2.58 ± 0.67; *p* < 0.001), Hamilton Anxiety Scale (16.75 ± 5.14 to 6.83 ± 3.79; *p* < 0.001), and Pittsburgh Sleep Quality Index (1.47 ± 0.99 to 0.82 ± 0.75; *p* < 0.02), indicating decreased sleep latency.

### 3.3. PBM in Parkinson’s Disease

Bullock-Saxton et al. [20] conducted a 4-week, randomized, double-blind, placebo-controlled study exploring the effect of combined transcranial and intra-oral photobiomodulation therapy on physical and cognitive outcome measures for people with Parkinson’s disease. Twenty-two participants received either sham and/or active laser photobiomodulation (904 nm, 60 mW/diode, 50 Hz) for 33 s to each of 21 points at the cranium and intra-orally, on one, two, or three times/week for 4 weeks. Two treatment phases were separated by a 4-week wash-out (Phase 2). The Montreal Cognitive Assessment was evaluated prior to treatment Phase 1 and at the end of treatment Phase 3. The Montreal Cognitive Assessment remained stable between the start and end of the study. No measures demonstrated statistically significant changes. With regular treatment, the spiral (writing) test and the dynamic step test were most sensitive to change in a positive direction; the nine-hole peg test demonstrated a minimum clinically important difference. A placebo effect was noted.

### 3.4. PBM in Sexual Disfunctions

Cassano et al. [21] conducted a secondary analysis of an 8-week, randomized, sham-controlled trial of transcranial photobiomodulation (t-PBM) for adults with major depressive disorder. The effects of t-PBM on sexual dysfunction were assessed in this study. A total of twenty participants underwent near-infrared (NIR) t-PBM (n = 9) or sham therapy (n = 11) twice a week for 8 weeks. Sexual desire, arousal, and orgasm were evaluated using the Systematic Assessment for Treatment-Emergent Effects—Specific Inquiry (SAFTEE-SI). The average improvement in sexual function (indicated by a decrease in the SAFTEE sex total score) was significantly greater in those receiving NIR-mode t-PBM compared to those receiving sham-mode, both in the overall sample (NIR [n = 9] − 2.55 ± 1.88 vs. sham [n = 11] − 0.45 ± 1.21; z = 2.548, *p* = 0.011) and among completers (NIR [n = 5] − 3.4 ± 1.95 vs. sham [n = 7] − 0.14 ± 1.21; z = 2.576, *p* = 0.010).

### 3.5. PBM in Traumatic Brain Injury

Costa Carneiro et al. [22] carried out a multidisciplinary clinical trial on 10 chronic adult patients (between 18 and 60 years old; an average age of 37.8 ± 10.2 years; both genders). The time after traumatic brain injury (TBI) ranged from 4 months to 4 years. Every patient received transcranial photobiomodulation therapy 3 times a week, for 6 weeks—cumulatively, 18 sessions, 30 min per session. Subjects were evaluated with neuropsychological tests (the Beck Depression Inventory, the Beck Anxiety Inventory, the Stroop test, the Trail Making Test, the Symbol Digit Test, the Ray Auditory Verbal Learning Test, the Complex Ray Figure, and the Verbal Fluency Test) and underwent transcranial Doppler (TD) to assess the peak systolic velocity (PSV) of the right and left middle cerebral arteries. The most common form of TBI was diffused axonal injury—DAI. The results of TD indicated that the participants affected for up to 8 months had better responses to the PBM than those with other types of lesions. Bilateral TD of MCA presented a significant increase in the PSV on the left side [*p* < 0.007]. None of the psychological assessments achieved statistical significance, but an improvement was noticed in relation to pre-PBM and post-PBM in the Ray Figure Test, Trail Making Test, the Verbal Fluency, test and the Ray Auditory Verbal Learning Test.

Naeser et al. [23] conducted a pilot, open-protocol study to evaluate the efficacy of transcranial PBM in the improvement of cognitive functions, particularly executive functions and verbal memory, in chronic mild traumatic brain injury patients. The time after TBI ranged from 10 months to 8 years (mean 38.2 months). Chronic cognitive problems ought to be presented for at least 6 months prior to screening evaluation. Eleven participants (26–62 years old; six males) received 18 sessions of transcranial PBM 870 nm wavelength, three times per week for 6 weeks. A significant linear trend was observed over time for the Stroop test for executive function, trial 3 inhibition (*p* = 0.004); Stroop, trial 4 inhibition switching (*p* = 0.003); the California Verbal Learning Test, total trials 1–5 (*p* = 0.003); and the California Verbal Learning Test, long-delay free recall (*p* = 0.006).

Justin E. Karr et al. [24] investigated whether photobiomodulation using red/near-infrared light applied transcranially via light-emitting diodes (LEDs) was linked to reduced symptoms and enhanced cognitive function in patients with chronic symptoms following mild traumatic brain injury. The study involved nine participants (three men, six women; ages 22–61) who underwent a 6-week intervention consisting of eighteen 40 min transcranial LED treatment sessions. Only two participants showed improvement in neuropsychological tests. On questionnaires, four reported enhanced cognition, five noted improved post-concussion symptoms, and three experienced better mood. Significant improvements in two or more areas were reported by four participants, with these improvements largely sustained at both follow-ups. However, a minority of participants self-reported symptom improvement, which may have been influenced by the intervention, changes in psychiatric medications, placebo effects, or other factors.

Maria Gabriela Figueiro Longo et al. [25] conducted a randomized, single-center, prospective, double-blind, placebo-controlled, parallel-group trial to assess the feasibility and safety of near-infrared low-level light therapy (LLLT) administered acutely after a moderate traumatic brain injury (TBI). The neuroreactivity to LLLT through quantitative magnetic resonance imaging metrics and neurocognitive assessment was examined. Sixty-eight patients with moderate traumatic brain injury were randomized to receive LLLT or sham therapy, and twenty-eight of them completed at least one LLLT session without any reported adverse events. Forty-three patients (twenty-two men [51.2%]; mean [SD] age, 50.49 [17.44] years]) completed the study with at least one magnetic resonance imaging scan: nineteen individuals in the LLLT group and twenty-four in the sham treatment group. Radial diffusivity (RD), mean diffusivity (MD), and fractional anisotropy (FA) showed significant time and treatment interaction at the 3-month time point (RD: 0.013; 95% CI, 0.006 to 0.019; *p* < 0.001; MD: 0.008; 95% CI, 0.001 to 0.015; *p* = 0.03; FA: −0.018; 95% CI, −0.026 to −0.010; *p* < 0.001). The LLLT group had lower Rivermead Post-Concussion Symptoms Questionnaire (RPQ) scores, but this effect did not reach statistical significance (time effect *p* = 0.39, treatment effect *p* = 0.61, and time × treatment effect *p* = 0.91). There were statistically significant differences observed in the late subacute stage in the magnetic resonance imaging-derived diffusion parameters of the white matter tracts between the sham- and light-treated groups, demonstrating neuroreactivity to LLLT.

Based on previous data, investigators from the Department of Neurosciences, University of California San Diego, hypothesized the neuroprotective effects of near-infrared therapy in patients with acute ischemic stroke.

Yair Lamp et al. [26] conducted a double-blind, randomized study NeuroThera Effectiveness and Safety Trial-1 (NEST-1) involving 120 ischemic stroke patients, in a randomized 2:1 ratio, with 79 patients in the active treatment group and 41 in the sham (placebo) control group, to evaluate the safety and preliminary effectiveness of the NeuroThera Laser System (infrared laser technology) in the ability to improve 90-day outcomes in ischemic stroke patients treated within 24 h of stroke onset.

In this study, more patients (70%) in the active treatment group had successful outcomes than did controls (51%), as measured prospectively on the bNIH (*p* = 0.035 stratified by severity and time to treatment; *p* = 0.048 stratified only by severity). Similarly, more patients (59%) had successful outcomes than did controls (44%), as measured at 90 days as a binary mRS score of 0 to 2 (*p* = 0.034 stratified by severity and time to treatment; *p* = 0.043 stratified only by severity). More patients in the active treatment group had successful outcomes than did controls, as measured by the change in mean NIHSS score from baseline to 90 days (*p* = 0.021 stratified by time to treatment) and the full mRS (“shift in Rankin”) score (*p* = 0.020 stratified by severity and time to treatment; *p* = 0.026 stratified only by severity). The prevalence odds ratio for bNIH was 1.40 (95% CI, 1.01 to 1.93) and for binary mRS was 1.38 (95% CI, 1.03 to 1.83), controlling for baseline severity. Similar results held for the Barthel Index and Glasgow Outcome Scale. Mortality rates and serious adverse events (SAEs) did not differ significantly (8.9% and 25.3% for active 9.8% and 36.6% for control, respectively, for mortality and SAEs).

The study indicated that infrared laser therapy has shown initial safety and effectiveness for the treatment of ischemic stroke in humans when initiated within 24 h of stroke onset.

Justin A. Zivin et al. [27] conducted a double-blind, randomized study (NEST-2) as the continuation of the previous findings. They compared transcranial laser therapy (TLT) to a sham control. In total, 331 patients received TLT and 327 received the sham; 120 (36.3%) in the TLT group achieved favorable outcomes versus 101 (30.9%) in the sham group (*p* = 0.094), with an odds ratio of 1.38 (95% CI, 0.95 to 2.00). Comparable results were seen for the other outcome measures. Post hoc analysis of patients with a baseline National Institutes of Health Stroke Scale score of <16 showed a favorable outcome at 90 days on the primary end point (*p* < 0.044). TLT within 24 h from stroke onset demonstrated safety but did not meet formal statistical significance for efficacy.

Based on Phase 2 results, the same group of investigators conducted another double-blind, sham-controlled, randomized clinical trial, Phase 3 (NEST-3) [28], but the study was terminated on recommendation by the Data Monitoring Committee after a futility analysis of 566 completed patients found no difference in the 90-day functional outcome as assessed by the modified Rankin scale [29].

### 3.6. PBM in Psychotic Disorders

Kheradmand et al. [30] conducted a randomized, double-blind, placebo-controlled clinical trial to evaluate the effect of PBM (810 nm wavelength) on cognitive impairment among patients with chronic schizophrenia. They enrolled 17 schizophrenic patients to the active treatment arm and 15 to the sham arm. Subjects were examined with the Mini Mental State Examination scale (MMSE) and Positive and Negative Syndrome Scale (PANSS). There was no significant difference in both MMSE and PANSS test scores in both groups after six sessions of PBM therapy and after 2 months of follow-up. The only one statistically significant score was in the depression/anxiety item in the PANSS (*p* = 0.028).

## 4. Discussion

Photobiomodulation is a novel, very promising method of treatment in psychiatric disturbances according to the data presented above. The majority of studies reported cognitive and affective improvement among patients who underwent the experimental procedure.

The biological effect of PBM is based on a photochemical mechanism. The NIR light is absorbed by mitochondrial chromophores, especially cytochrome C oxidase (CCO). This is a transmembrane protein complex and the last enzyme in the respiratory electron transport chain located in the internal mitochondrial membrane. PBM initiates the respiratory chain reaction, reducing oxidative stress and increasing transmembrane potential, and thus neural reactivity.

In case of intranasal near-infrared stimulation, the target is the default mode network (DMN), a resting-state network responsible for maintaining the basic activity of cognitive functions during resting periods as well as the metabolic balance of the CNS. The default mode network (DMN) is a large-scale brain network composed of the dorsal medial prefrontal cortex, posterior cingulate cortex, precuneus, and angular gyrus. Studies conducted so far [31] indicate impaired structural and functional connectivity within the DMN in the course of Alzheimer’s disease and Parkinson’s disease. Structures belonging to the DMN coincide with the regions of deposition of amyloid plaques Aβ-40/42. Numerous clinical trials have shown that infrared stimulation causes the synchronization of connections within the DMN, especially between the posterior cingulate cortex and lateral parietal cortex, responsible for episodic memory (the most often reduced cognitive domain in AD).

The conducted analyses of PBM’s effects on selected disorders shows that this method has the greatest impact on cognitive functions, particularly memory. In cases of dementia, depression, and anxiety disorders, it specifically affects memory consolidation and recall based on memory traces, including delayed recall. Additionally, in dementia due to Alzheimer’s disease, it has a positive impact on maintaining sustained attention and alertness, improves executive functions, and shortens reaction time. This is reflected in specific brain regions being stimulated and PBM’s described effects on them. Among patients with traumatic brain injury, significant positive changes were observed in verbal memory, recall, and executive functions.

No effect of PBM was observed on either cognitive functions or motor symptoms in Parkinson’s disease. The authors explain this by the variability of symptoms of Parkinson’s disease in individuals, necessitating a more individualized adjustment of stimulation sites and doses for each patient.

No positive effects of PBM on positive or negative symptoms in psychotic disorders, including cognitive disorders, were observed either.

The antidepressive and anti-anxiety effect of PBM was also evaluated in an experimental mouse model of chronic restraint stress [32]. It was found that near-infrared stimulation significantly increased 5-HT and decreased NO levels in the prefrontal cortex and hippocampus. Suppression of Nitric Oxide Synthase (NOS) activity could be considered as one of the hypothetic molecular mechanisms of PBM. Another significant aspect in the development of depressive disorders is the involvement of inflammatory factors. Psychosocial stress is a well-established and consistent predictor of depression in humans and is commonly used to induce depressive-like behavior in laboratory animals. Consequently, the finding that psychosocial stress in a lab setting can trigger an inflammatory response in humans was a significant advance in connecting inflammation with depression [33]. Psychosocial stress can also cause microglia to adopt a pro-inflammatory macrophage M1 phenotype, releasing CC-chemokine ligand 2 (CCL2), which attracts activated myeloid cells to the brain through a cellular pathway. Once in the brain, these activated M1 macrophages can sustain central inflammatory responses [34]. Studies have demonstrated the anti-inflammatory effects of photobiomodulation. One of the properties of PBM is its ability to change the phenotype of activated macrophage cell lines from the pro-inflammatory M1 phenotype to the anti-inflammatory M2 phenotype [35]. Detailed connections between the mechanisms of NIR and disorders are presented in Table 1.

It has also been proven that a number of psychiatric disorders are associated with abnormal cerebral blood flow. Improvement in systolic/diastolic flow velocity is another investigated positive effect of PBM treatment, through altering the elasticity and resistance of the cerebrovascular wall of the blood arteries [36].

Despite these proven facts, conducted studies vary widely in methodology. First of all, researchers use different wavelengths, in the scope of 823 nm to 1087 nm, which results in diverse penetration of the tissues, due to various depths of radiation. Studies on dementia have used wavelengths around 1064 nm, while studies on depression and anxiety have used wavelengths like 823 nm and 945 nm. Another important issue is the target of photobiomodulation waves and the place of stimulation. In most cases, transcranial PBM is performed, but intranasal or indirect stimulation through another part of the human body is also performed, which could influence the final results. Not everyone experiences the same level of benefit from NIR therapy, and its effects can vary based on individual conditions and treatment protocols. Moreover, the studies differ with regard to the scales and surveys used in patients’ assessment.

All of these differences cause disturbances in the interpretation of findings and make them frequently incomparable to one another.

An additional important disadvantage of the studies mentioned above is the small number of enrolled subjects, which is not representative enough. Despite that, it is noteworthy that researchers achieved statistical significance, which indicates the strong influence of PBM on the central nervous system. Although short-term benefits are well documented, comprehensive long-term studies on the effects and potential risks of NIR therapy are still limited.

For these reasons, further studies are needed, conducted on a greater number and more homogenous groups of individuals, with a higher level of methodology unification.

## 5. Conclusions

Photobiomodulation, due to its mechanism of action, different from available drugs for mental disorders, can be an independent and additive method of treatment. Due to the prevalence of mental problems in the world, it is also worth noting the low cost of equipment and ease of access to this form of therapy.

## Figures and Tables

**Figure 1 neurosci-06-00026-f001:**
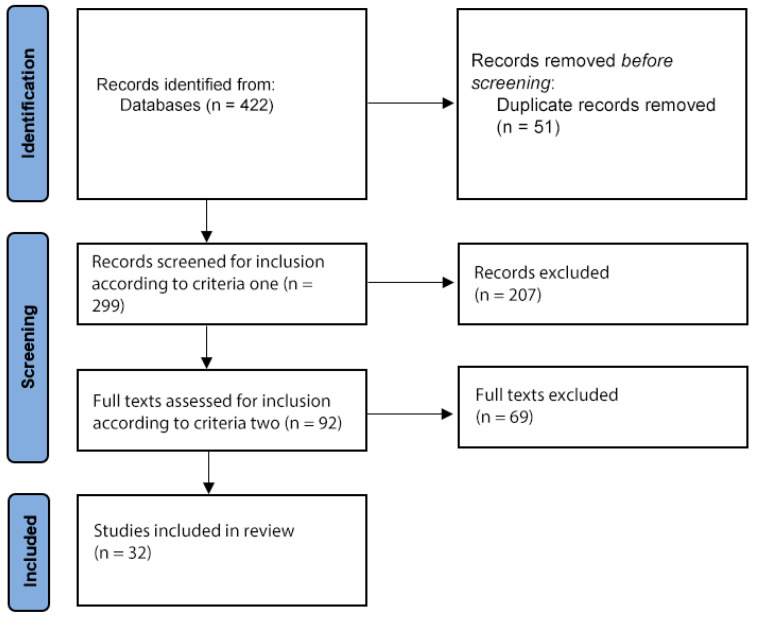
Flow diagram of study selection.

**Table 1 neurosci-06-00026-t001:** Detailed connections between the mechanisms of NIR and disorders.

Psychiatric Disorder	Mechanism
Alzhemer’s disease dementia	• Synchronization of connections within the DMN, especially between the posterior cingulate cortex and lateral parietal cortex; • Reducing oxidative stress and increasing transmembrane potential; • Increasing power spectral density for gamma and beta frequency bands in open-eye EEG.
Depression and anxiety	• Increasing 5-HT and decreasing NO levels in the prefrontal cortex and hippocampus; • Suppression of nitric oxide synthase (NOS) activity; • Changing the phenotype of activated macrophage cell lines from the pro-inflammatory M1 phenotype to the anti-inflammatory M2 phenotype.
Sexual dysfunctions	• Increasing 5-HT and decreasing NO levels in the prefrontal cortex and hippocampus; • Suppression of NOS activity.
Traumatic brain injury	• Reducting oxidative stress and increasing trans membrane potential; • Increase in the peak systolic velocity (PSV) of middle cerebral arteries.

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
