# Peer review of "Near-Infrared Stimulation in Psychiatry Disorders: A Systematic Review of Efficacy and Biological Mechanisms"

_neurosci, 2025, doi:10.3390/neurosci6010026_

Round 1
Reviewer 1 Report
Comments and Suggestions for Authors
The review by Woźniak et al. is interesting. However, several points need to be addressed:
-
The review covers the effect of photobiomodulation (PBM) on mental disorders, which is a good idea. However, the conclusions section is empty—it seems the authors forgot to fill in this section.
-
The discussion section is quite short, and it would be helpful to elaborate on the similarities and differences in the effects of PBM on various diseases.
-
It is important to mention any mechanistic studies that connect photobiomodulation with pathology or symptom reduction. For example, the authors mentioned mitochondria—are these effects seen in specific cells? How does this affect mood, and which areas of the brain are involved?
-
The introduction section lacks a more detailed description of the method. How is PBM performed? What are the risks and advantages? What was the original hypothesis of the research, and is it proven or not?
-
It might be a good idea to include a summary table by disease and effects, so researchers can easily refer to it.
-
Overall, the connectivity between sections is lacking. It would be helpful if the authors explained why they chose these specific pathologies and not others, such as epilepsy, multiple sclerosis, or other physiological or mental diseases. Currently, it is unclear why they focused on the diseases included in the review.
Author Response
Reviewer 1: To whom it may concern,
We are grateful for the time and effort the Reviewer dedicated to reviewing our manuscript.
The Reviewer's comments helped us identify essential aspects of the research that required refinement. We believe these revisions will strengthen the overall work.
Specifically:
- A new "Conclusions" paragraph has been added, summarizing the key findings in a concise manner.
- In the "Discussion" section, a new paragraph was included that outlines the similarities and differences between the effects of PBM in selected diseases.
- A more detailed analysis of the mechanisms of PBM action was provided, especially the target areas/cells of action.
- The exact aim and hypothesis of the study were defined in the introduction.
- New Table summarizing the biological mechanisms of near-infrared stimulation effectiveness
- The reasons for focusing on specific selected disorders were specified.
Once again, thank you for your invaluable support.
Reviewer 2 Report
Comments and Suggestions for Authors
The manuscript explores the therapeutic potential of near-infrared stimulation (NIR) for psychiatric disorders, focusing on its mechanisms of action and evidence from preclinical and clinical studies. It highlights the non-invasive nature of photobiomodulation (PBM), its safety profile, and its emerging role as an alternative to traditional pharmacological treatments. Disorders such as dementia, depression, anxiety, Parkinson’s disease, and traumatic brain injury are reviewed for their responsiveness to PBM.
1. Keywords like "psychiatry" and "neuromodulation" are overly broad and do not reflect the manuscript’s specific focus.
2. Although the mechanisms of PBM are discussed, the connection between these mechanisms and observed clinical outcomes is not sufficiently detailed.
3. The manuscript does not critically evaluate the statistical robustness of the included studies, leaving the reliability of the findings uncertain.
4. The manuscript contains grammatical errors (e.g., “do not interact with other drugs”), inconsistent abbreviations, and awkward phrasing.
5. The figures, particularly the PRISMA diagram, lack clarity and should be revised for better readability.
6. Provide more detailed connections between the mechanisms of NIR (e.g., mitochondrial activity) and its clinical effects across disorders.
7. What are the specific mechanisms by which PBM might differentially affect disorders such as depression versus neurodegenerative conditions?
Author Response
Revewer 2: To whom it may concern,
We are grateful for the time and effort the Reviewer dedicated to reviewing our manuscript.
The Reviewer's comments helped us identify essential aspects of the research that required refinement. We believe these revisions will strengthen the overall work.
Specifically:
- A new keywords are added
- In the "Discussion" section, a new paragraph was included that outlines the common effects of PBM in various diseases. A more detailed analysis of the mechanisms of PBM action was provided.
- The suggested language corrections have been implemented in the indicated sections (point 4)
- PRISMA diagram is implemented directly from PRISMA- statement webside.
The PRISMA 2020 flow diagram templates are distributed in accordance with the terms of the Creative Commons Attribution (CC BY 4.0) license. To view a copy of this license, visit https://creativecommons.org/licenses/by/4.0/ - New Table summarizing the biological mechanisms of near-infrared stimulation effectiveness is added (point 6 and 7)
Round 2
Reviewer 1 Report
Comments and Suggestions for Authors
Thank you for addressing my comments.
Reviewer 2 Report
Comments and Suggestions for Authors
Authors have addressed my all queries.